# The Integrated Scheduling Optimization for Container Handling by Using Driverless Electric Truck in Automated Container Terminal

Cheng Hong [1], Yufang Guo [2], Yuhong Wang [1,*] and Tingting Li [1]

[1]   Faculty of Maritime and Transportation, Ningbo University, Ningbo 315211, China
[2]   Ningbo Shunyu Infrared Technology Co., Ltd., Ningbo 315400, China
*   Correspondence: myuhongwang@nbu.edu.cn

**Abstract:** With the increasing maturity of automatic driving technology, the commercial value of driverless container trucks has been gradually excavated. Compared with social roads, the internal roads in the port area have certain practicing advantages. By taking into account the operational characteristics of the driverless electric container truck and the coordination of quay and yard cranes, this paper aims to explore the configuration and optimized scheduling model of the driverless electric container truck with the objective of minimizing overall energy consumption. The results show that the optimized allocation and scheduling of driverless electric trucks can minimize the total energy consumption of terminal operation without delaying the shipping schedule, and has obvious advantages over traditional manual driving diesel trucks and Automated Guided Vehicles in terms of operation efficiency, economy, and sociality. The results can also provide certain decision-making reference for the selection of horizontal transportation equipment and collaborative scheduling of multi-type loading and unloading equipment resources of container terminal operators.

**Keywords:** automated container terminal; driverless truck; resource allocation; scheduling optimization

## 1. Introduction

Automated Container Terminal (ACT), with the automated container handling process in quayside loading and discharging, horizontal movement and yard storage, becomes an inevitable tendency of future port development due to its advantages in reducing handling costs, improving operational efficiency, and prompting its reputation of green performance [1,2] Port of Rotterdam, Hamburg and several other leading hubs played a leading role in developing ACT since the 1990s. Over the past decades, continuous effort has been given to the tradeoff between capital costs of automation facilities and terminal overall performance, while the application of autonomous driving technology on the container horizontal transportation, or even a wider freight transportation system, serves as a pioneer example in leading this tendency [3].

Traditionally, Automated Guided Vehicles (AGV) have been widely deployed in many large container terminals worldwide due to their advantages in increased safety, productivity and significant labor cost savings [4], and even more environmental benefits if they use eco-friendly technology such as battery energy driven [5,6]. On the other hand, driving automation technology becomes more mature and attractive [7] evidenced by a number of literatures for its contribution in transport safety [8,9], driving experience [10], traffic efficiency [11], and economic achievement [12]. The electricity-powered driverless trucks (DET), when comparing with traditional AGV, are able maintain similar advantages but with far less initial capital requirement and there is no need to bury transponders (also known as magnetic nails) underneath road surface in advance [13]. It means that the DET, by taking its advantages in navigation and positioning technology, will have more flexibility to fit for different shape and layout of terminal infrastructures, which is particularly helpful

when converting manual horizontal container movement to the automated one [14]. As a result, a form of "Double Trolley Quay Crane (DTQC) + Driverless Electric Truck (DET) + Automatic Rail Mounted Gantry (ARMG)" system becomes popular in most recent terminal automation projects such as the port of Zhuhai (2018), Rotterdam's Maasvlakets (2019), Ningbo (2020) and Tianjin (2022). However, the resource allocation and scheduling of such handling system require coordination among different components, to improve the overall operating performance, which becomes an optimization problem. Such optimization makes sense because nowadays with the development of Automated Container Terminal, the automatic decision about the operation plan in the port becomes more and more important. This work provides a convenient tool for the container handling plan, which is helpful to improve the efficiency of the port operation.

There has been abundant research made on the configuration and scheduling of various types of equipment in the terminal, no matter the traditional manual operation or in automated format. Jia et al. [15] studied a vessel traffic scheduling problem considering the inter-shipping line equity issue. Considering the uncertainty of the service times of feeders, Jia et al. [16] studied the problem of how to allocate berths to deep-sea vessels and schedule arrivals of feeders for congestion mitigation at a container port. Kim and Park [17] investigated the quay crane resource allocation issues by considering the minimum safety distance requirement and the task priorities of loading and discharging. Bierwirth and Meisel [18] emphasized the interference constraints between quay cranes in the scheduling model, while Zhang et al. [19] optimized the quay crane dispatching by paying attention to the longitudinal stability of the ship during loading and unloading. In regard to the optimization of the quay crane operation, Goodchild and Daganzo [20,21], Zeng et al. [22], Zhang and Han [23] adopted the synchronous loading and unloading operation mode for developing the quay crane configuration and scheduling model, which aimed to minimize the quay crane operation time and improve efficiency through optimizing the loading and unloading sequence of container vessels.

Attention has also been given to the horizontal movement of containers connecting the quayside and storage yard in recent years. For example, in regard to the scheduling of horizontal movement activities, Legato et al. [24] firstly addressed the potential impact of the inconsistency of the quay crane loading and discharging; Ma and Hu [25] noticed the uncertainty caused by traffic congestion in the container storage yard; Zhang et al. [26] suggested to take into account the Automated Guided Vehicle (AGV) endurance; additionally, Adamo et al. [27] further added the varying speeds of AGVs in each path as another influential factor. Most recently, Hu [28] constructed an AGV configuration and scheduling model with the pool strategy to maximize the utilization of AGV. Liu and Ge [29] investigated the quay crane assignment issues from the perspective of minimizing $CO_2$ emission during an unloading process of containers from quay cranes to AGVs. Their research finding suggested a positive correlation between quay crane resources and horizontal movement vehicles that the optimal number of quay cranes increases with the expected arrival rate of AGVs.

As stated by He et al. [30] that the scheduling of quay cranes and AGVs are actually two highly related production decision-making issues during the process of loading and unloading operations. The former determines the operating time of ships, while the latter affects the loading/unloading time of the quay cranes and automatic rail mounted gantry. As a result, more and more scholars shift their attention to the coordinated scheduling of various equipment in the terminal operations. Kizilay et al. [31] established the safety distance and interference constraint model for joint optimization of quay crane allocation and scheduling, yard crane allocation and scheduling, yard location allocation and yard truck allocation and scheduling, so as to minimize the rotation time of ships and improve the throughput of the terminal. Peng et al. [32] quantified the impact of equipment configuration on the total carbon emissions in terminal operations, established a simulation model based on a complex queuing network, and thereby optimized the ratio of the quay crane, yard crane and AGV. [33–37] studied the collaboration optimization of the berth

allocation and quay crane scheduling problem. Yin et al. [38] studied the quay cranes and shuttle vehicles simultaneous scheduling problem considering limited apron buffer capacity. Cahyono et al. [39] studied the simultaneous allocation and scheduling of quay cranes, yard cranes, and trucks in dynamical integrated container terminal operations. Çağatay Iris et al. [40–42] presented the flexible containership loading problem for seaport container terminals, in which the integrated management of ship loading operations, including operational stowage planning, load sequencing, planning of the equipment to use and their scheduling, is addressed. Gao and Ge [43] studied a multiple yard crane scheduling problem, aiming to minimize both total longitudinal distance of yard cranes and total waiting time of internal and external trucks.

The existing literatures provide a concrete theoretical foundation for optimized allocation of terminal resources which, in consequence, lead to significant improvement of container terminal productivity and efficiency. However, as mentioned earlier, DET exhibits a completely different operational features to traditional AGV or manual truck mode due its significant saving of initial investment, great flexibility in moving paths and decoupling operation in storage yard. In addition, in relation to the operation and scheduling of container handling system, currently, the integrated scheduling of the overall container handling system which consists of vertical loading and discharging, and the horizontal movement has not been detailed discussed yet, but such an operation mode is more advanced and has a wide application prospect in ACTs.

Therefore, this work intends to make some improvements related to the abovementioned issues. An integrated operation mode, "Double Trolley Quay Crane (DTQC) + Driverless Electric Truck (DET)" is taken into account, in which synchronous loading and unloading operation mode is used to dispatch quay crane, and the pool strategy is used to dispatch DET to respond to the transportation service demand of the quay crane and yard crane, so as to achieve higher service quality with less equipment resources. Operationally, DET will be utilized to replace the traditional container truck or AGV. Focusing on this mode, the corresponding integrated optimization model is developed, aiming at optimizing the scheduling of the overall container handling system. By doing so, an integrated resource allocation strategy could be obtained to support the operation of the recently developed container terminal automation projects and future designs.

This work has two aspects of contributions: first, an integrated scheduling optimization model for the DTQC-DET system is proposed, providing an effective tool of synthetical plan for the operation in ACT. Second, the Driverless Electric Truck is considered and compared with traditional transportation mediums, indicating its potential of the application in ACT.

The remaining of this paper is organized as follows: Section 2 clarifies the problem and figures out the structure of how to solve the problem. Section 3 develops three optimized models for (1) container loading and discharging sequence; (2) quayside cranes allocation and scheduling; and (3) horizontal container movement vehicle allocation and scheduling, which consists of the process of ACT operations. The effectiveness of these models are validated in Section 4 with the application of an example case. In addition, a further comparison between DET and the traditional horizontal transport vehicles (i.e., AGV and manual driving diesel truck) is made to investigate the operational and economical features of DET. Finally, conclusions are drawn in Section 5.

## 2. Problem Statement

Nowadays, the "DTQC + DET + ARMG" is currently the most widely utilized operation mode in ACT operations. In so doing, the quayside cranes and gantry crane in the yard are responsible for the loading and discharging of containers in berthing position and storage yard, respectively, while the DETs transport containers back and forth between the front of the quayside and the yard area. By taking the unloading process as an example, the imported container is lifted by the main trolley of the DTQC to the transfer platform, and then further shift the DETs by the gantry trolley. The DETs then transport these containers

to the buffer bracket in front of the designated import yard based on the provided stacking plan. The ARMG on the seaside of the import yard picks up the imported containers and places them in the appropriate stacking space of the yard for temporary storage. The loading process is opposite to the unloading process. The specific operation process is shown in Figure 1.

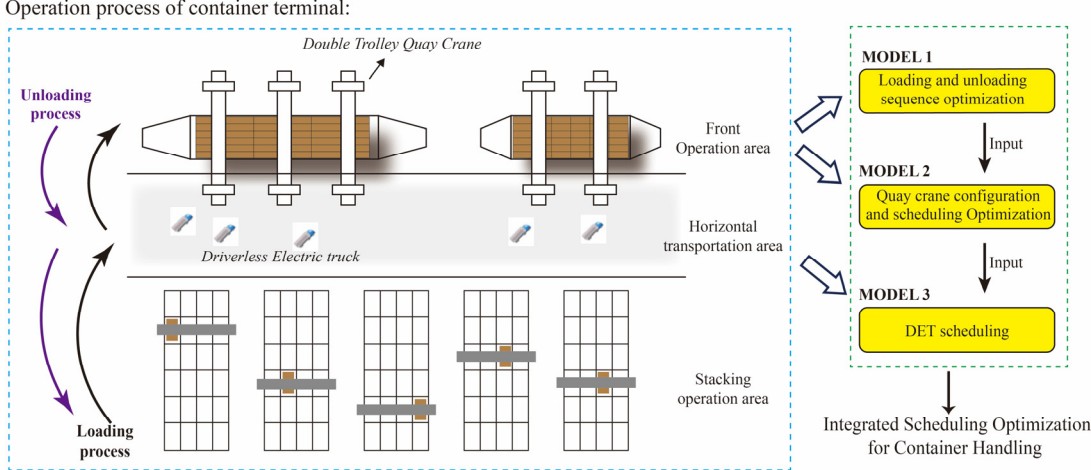

**Figure 1.** Operation process of container terminal and the corresponding integrated models.

According to the above discussions, for the "DTQC + DET + ARMG" system, the quay crane is dispatched with the simultaneous loading and unloading operation mode of the same bay, so as to complete the loading and unloading of all containers within the specified time based on the shipping schedule. Additionally, the DET is dispatched according to the quay crane operation plan. Therefore, in order to achieve the integrated optimized scheduling plan of this operational mode, the overall performance improvement has been divided into the following three steps: (1) optimizing the loading and unloading plan of the containers for optimized quay crane resource allocation; (2) upon the loading and unloading plan, determine the optimized container transportation plan of DETs; (3) based on the optimized quay crane operation plan, design the corresponding scheduling strategy of DTQC and ARMG. Each step corresponds to one optimization model.

## 3. Mathematical Model

### 3.1. Assumptions

In order to develop the three models mentioned above, essential assumptions are made as follows:

(1) The quay crane can only operate one container each time, and the size of the container loaded and unloaded is the same, which is a 40 ft standard container;

(2) In relation to the loading and unloading process in the bay, considering that the loading operation for one row has to start after the unloading operations of the containers in this row all finish, the loading/unloading sequences related to tiers are not considered. Instead, the parameters representing one whole row are used, and we focus on the row sequences of the loading and unloading operations.

(3) All quay cranes move on the same track with the same operation efficiency and energy consumption;

(4) The import container area is separated from the export container area in the yard, the capacity in the container area is sufficient, and the service capacity can meet the task requirements of all containers;

(5) There is a buffer stand in front of each container area, and there is one and only one ARMG working on the seaside of the container area, and all the ARMGs have the same work efficiency, energy consumption and other performance;

(6)  Each equipment has the same operating capability, and ARMG and DETs can only operate one container at a time;

(7)  Failures during the operation of the quay crane are not considered.

### 3.2. Notations of the Proposed Models

The defined nations for the three models are shown in the Appendix A, from Tables A1–A3 correspondingly.

### 3.3. Optimization Models
3.3.1. MODEL 1: Optimization of Loading and Unloading Sequence in Bay

Equations (1)–(10) is the mathematical model for the Optimization of Loading and Unloading Sequence in Bay.

$$\min \max\left\{e_{br}^{l}\right\}, \forall b \in B, \forall r \in R \tag{1}$$

Subject to

$$\sum_{r' \in R} x_{rr'} - \sum_{r' \in R} x_{r'r} = 0, \forall r \in R \tag{2}$$

$$\sum_{r' \in R} y_{rr'} - \sum_{r' \in R} y_{r'r} = 0, \forall r \in R \tag{3}$$

$$s_{br}^{l} + T_1 N_{br}^{l} \leq e_{br}^{l}, \forall r \in R \tag{4}$$

$$s_{br}^{u} + T_1 N_{br}^{u} \leq e_{br}^{u}, \forall r \in R \tag{5}$$

$$s_{br'}^{u} - (1 - x_{r'r}) \cdot M \leq e_{br}^{u} \leq s_{br'}^{u} + (1 - x_{r'r}) \cdot M, \ \forall r, r' \in R \tag{6}$$

$$s_{br'}^{l} - (1 - y_{r'r}) \cdot M \leq e_{br}^{l} \leq s_{br'}^{l} + (1 - y_{r'r}) \cdot M, \ \forall r, r' \in R \tag{7}$$

$$s_{br}^{l} - e_{br}^{u} \geq 0, \forall r \in R \tag{8}$$

$$T_{bq} \geq 0, \forall b \in B, \forall q \in Q \tag{9}$$

$$x_{rr'}, y_{rr'} \in \{0, 1\}, \forall r, r' \in R \tag{10}$$

Equation (1) indicates the objective function. Within each bay, the total quay crane operation time is the time for the quay crane to complete the last container operation. Optimizing the sequence of loading and unloading in the bay is to minimize the loading and unloading operation time. Considering that the total quay crane operation time is the time for the quay crane to complete the last container operation, which could be presented by the maximum of the loading time for the last container. Therefore, the objective function becomes a min–max form, as shown by Equation (1).

Equations (2) and (3) show that there is only one pre-operation and one post-operation when the quay crane carries out the unloading or loading task of each container.

Equations (4) and (5) set up the operation time constraint of each container, which is that the time interval is not less than the loading and unloading time of all the import containers in the loading operations.

Equation (6) defines the relationship between the unloading sequences and the starting/ending times. If $x_{rr'} = 1$, containers in row $r'$ will be unloaded after row $r$; therefore, the starting time of row $r'$ should be equal to the end time of row $r$, otherwise, if $x_{rr'} = 0$, row $r$ and row $r'$ are not consecutive, so that a larger arbitrary number "M" is added to cancel

such constraint. Equation (7) uses the same way as Equation (6) to define the relationship between the loading sequences and the corresponding starting/ending times.

Equation (8) indicates the operation order, which is that it is necessary to unload the import container first, and then carry out the loading operation of the export container.

Equations (9) and (10) define the variable type and value range, respectively.

### 3.3.2. MODEL 2: Optimization of Quay Crane Configuration and Scheduling

The model for the optimization of quay crane configuration and scheduling is represented by Equations (11)–(19)

$$\min W_1 \times \sum_{b\in B}\sum_{q\in Q} x_{bq}T_b + W_2 \times \sum_{b\in B}\sum_{b'\in B}\sum_{q\in Q} |l_{b'} - l_b| \times z_{bb'q}T_2 + W_3 \times \sum_{b\in B}\sum_{b'\in B}\sum_{q\in Q} z_{bb'q}T_{bb'} \quad (11)$$

Subject to

$$\sum_{q\in Q} x_{bq} = 1, \forall b \in B \quad (12)$$

$$T_{bq} + \frac{l_s}{T_2} \leq T_{b'q'} - T_{b'}, \forall b, b' \in \psi, \forall q, q' \in Q \quad (13)$$

$$\sum_{s\in B} x_{sq} = 1, \forall q \in Q \quad (14)$$

$$\sum_{e\in B} x_{eq} = 1, \forall q \in Q \quad (15)$$

$$\sum_{b'\in B} z_{bb'q} - \sum_{b'\in B} z_{b'bq} = 0, \forall b \in B, \forall q \in Q \quad (16)$$

$$T_{bq} + |l_{b'} - l_b| \times T_2 + T_{b'} + T_{bb'} = T_{b'q}, \forall b, b' \in I, \forall q \in Q \quad (17)$$

$$\sum_{b\in B} x_{bq}T_b + \sum_{b\in B}\sum_{b'\in B} |l_{b'} - l_b| \times z_{bb'q}T_2 + \sum_{b\in B}\sum_{b'\in B} z_{bb'q}T_{bb'} \leq T_F, \forall q \in Q \quad (18)$$

$$x_{bq}, z_{bb'q} \in \{1, 0\}, \forall b, b' \in B, \forall q \in Q \quad (19)$$

Equation (11) defines the objective function, which is the minimum total energy consumption of quay crane operation. In Equation (11), the three terms are the total operation energy consumption, total moving energy consumption and total waiting energy consumption caused by interference, respectively.

Equation (12) means one quay crane can only operate one bay at a time.

Equation (13) is proposed to ensure the safety of operation, which is that the safe distance between two quay cranes must be kept at at least one bay. Therefore, if bay $b$ and bay $b'$ cannot be operated at the same time, quay crane $q$ can only start the operation after completing the loading and unloading operation in bay $b$ and moving to a safe distance.

Equations (14) and (15) show that there is only one starting bay and one ending bay for each quay crane.

Equation (16) means that there is and only one bay before and after when the quay crane is working at a certain bay.

Equation (17) is the time constraint for continuous operation of two bays. The same quay crane can only load and unload the next bay after completing all the loading and unloading tasks in one bay.

Equation (18) defines the total time constraint. The total completion time of all quay cranes is less than the specified completion time of loading and unloading in port.

Equation (19) defines the value range, indicating that the decision variables are binaries.

### 3.3.3. MODEL 3: Optimization of DET Scheduling

Model 3 is proposed for the optimization of DET scheduling, which is shown from Equation (20) to Equation (51).

$$\min W_4 \times \sum_{n \in N} \sum_{q \in Q} \left( T^r_{nq2} - T^p_{nq2} \right) + W_5 \times \sum_{n \in N} \sum_{a \in A} \sum_{q \in Q} \sum_{q' \in Q} y_{na} \left( T^{qU}_{na} + T^{qL}_{na} + T^{Uq'}_{na} + T^{Lq'}_{na} \right)$$
$$+ W_6 \times \left( \sum_{n \in N} \sum_{n' \in N} \sum_{a \in A} z_{nn'a} T_{nn'a} + 2 \sum_{a \in A} u_a T^E_a \right) + W_7 \times \left( \sum_{n \in N} \sum_{q \in Q} \sum_{a \in A} w_{naq} + \sum_{n \in N} \sum_{c \in C} \sum_{a \in A} w_{nac} \right) \tag{20}$$

Subject to

$$\sum_{a \in A} y_{na} = 1, \forall n \in N \tag{21}$$

$$\sum_{n' \in N} z_{nn'a} - \sum_{n' \in N} z_{n'na} = 0, \forall n \in N, \forall a \in A \tag{22}$$

$$\sum_{s \in N} y_{sa} = 1, \forall a \in A \tag{23}$$

$$\sum_{e \in N} y_{ea} = 1, \forall a \in A \tag{24}$$

$$T^r_{nq1} \le T^p_{nq1}, \forall n \in N \tag{25}$$

$$T^E_{nq2} = T^p_{nq1}, \forall n \in U, \forall q \in Q \tag{26}$$

$$T^L_{nq2} = T^p_{nq1} + p \times T_1, \forall n \in U, \forall q \in Q \tag{27}$$

$$T^E_{nq2} = T^p_{nq1} - p \times T_1, \forall n \in L, \forall q \in Q \tag{28}$$

$$T^L_{nq2} = T^p_{nq1}, \forall n \in L, \forall q \in Q \tag{29}$$

$$T^E_{nq2} \le T^r_{nq2} - \frac{1}{2}T_3 \le T^L_{nq2}, \forall n \in U, \forall q \in Q \tag{30}$$

$$T^E_{nq2} \le T^r_{nq2} + \frac{1}{2}T_3 \le T^L_{nq2}, \forall n \in L, \forall q \in Q \tag{31}$$

$$T^r_{nq2} = \max\left\{ T^p_{nq2}, T^q_{na} \right\}, \forall n \in U, \forall q \in Q, \forall a \in A \tag{32}$$

$$T^r_{nq2} = \max\left\{ T^p_{nq2}, T^L_{na} + T^{Lq}_{na} \right\}, \forall n \in L, \forall q \in Q, \forall a \in A \tag{33}$$

$$T^p_{n'q2} = T^r_{nq2} + T_3, \forall n, n' \in N, \forall q \in Q \tag{34}$$

$$T^{EU}_{na} = T^U_{nc} - p_2 \times T_4, \forall n \in U, \forall c \in C \tag{35}$$

$$T^{LU}_{na} = T^U_{nc}, \forall n \in U, \forall c \in C \tag{36}$$

$$T^{EL}_{na} = T^L_{nc}, \forall n \in L, \forall c \in C \tag{37}$$

$$T^{LL}_{na} = T^L_{nc} + p_2 \times T_4, \forall n \in L, \forall c \in C \tag{38}$$

$$T_{na}^{EU} \leq T_{na}^q + T_{na}^{qU} \leq T_{na}^{LU}, \forall n \in U, \forall a \in A, q \in Q \tag{39}$$

$$T_{na}^{EL} \leq T_{na}^L \leq T_{na}^{LL}, \forall n \in L, \forall a \in A, q \in Q \tag{40}$$

$$w_{naq} = \max\left\{T_{nq2}^p - T_{na}^q, 0\right\}, \forall n \in U, a \in A, q \in Q \tag{41}$$

$$w_{naq} = \max\left\{T_{nq2}^p - T_{na}^L - T_{na}^{Lq}, 0\right\}, \forall n \in L, a \in A, q \in Q \tag{42}$$

$$w_{nac} = \max\left\{T_{na}^{EU} - T_{na}^q - T_{na}^{qU}, 0\right\}, \forall n \in U, a \in A, q \in Q \tag{43}$$

$$w_{nac} = \max\left\{T_{na}^{EL} - T_{na}^L, 0\right\}, \forall n \in L, a \in A, q \in Q \tag{44}$$

$$T_{na}^q + T_{na}^{qU} + w_{nac} + T_{na}^{Uq'} + w_{n'aq} + 2T_a^E \times u_a = T_{n'a}^{q'}, \forall n, n' \in U \tag{45}$$

$$T_{na}^q + T_{na}^{qU} + w_{nac} + T_a^{UL} + w_{n'ac'} + 2T_a^E \times u_a = T_{n'a}^L, \forall n \in U, \forall n' \in L \tag{46}$$

$$T_{na}^L + T_{na}^{Lq} + w_{naq} + T_a^{qq'} + w_{n'aq'} + 2T_a^E \times u_a = T_{n'a}^{q'}, \forall n \in L, \forall n' \in U \tag{47}$$

$$T_{na}^L + T_{na}^{Lq} + w_{naq} + T_{n'a}^{qL} + w_{n'ac} + 2T_a^E \times u_a = T_{n'a}^L, \forall n, n' \in L \tag{48}$$

$$y_{na}, z_{nn'a}, u_a \in \{0, 1\}, \forall n, n' \in N, \forall a \in A \tag{49}$$

$$T_{na}^q \geq 0, \forall n \in N, \forall a \in A, \forall q \in Q \tag{50}$$

$$T_{na}^L \geq 0, \forall n \in N, \forall a \in A \tag{51}$$

Equation (20) defines the objective function, which is the minimum energy consumption in the dispatching process of DET. The four items in Equation (21) are energy consumption of gantry trolley waiting for driverless electric container truck, full-load energy consumption of driverless electric container truck, no-load energy consumption of driverless electric container truck, and waiting energy consumption in the buffer area in the quayside and yard, respectively.

Equation (21) means that each DET transports only one container at a time. Equation (22) indicates that DETs are continuous when carrying out container transportation tasks. Each DET has and only has one pre task and one post task. Equations (23) and (24) show that each DET has and only has one start task and one end task.

Equation (25) sets up the operation time constraint of a certain container. It is necessary to ensure that all container loading and unloading tasks can be completed within the specified completion time of loading and unloading in port, that is, the main trolley of dual-trolley quay crane will not be delayed, and the actual operation time shall not be later than the planned operation time.

Equations (26) and (27) indicate the earliest/latest time for the gantry trolley to pick up the import container on the transfer platform. Equations (28)–(30) indicate the earliest/latest time for gantry trolley to put down the export container on the transfer platform. Equation (30) shows the actual operation time of the gantry trolley of the dual-trolley quay crane in handling import containers on the transfer platform, while the operation time in handling the export container is shown by Equation (31). Equations (32) and (33) are proposed to ensure that the DET has reached the quayside during the actual operation of

the gantry trolley. Equation (34) shows that the planned operation time of gantry trolley shall be updated according to the actual operation situation.

Equations (35) and (36) represent the earliest/latest time for the DET to put down containers on the buffer bracket in front of the import container area. Similarly, Equations (37) and (38) represent the earliest/latest time for the export container to pick up from the buffer bracket in front of the export container area. Equation (39) defines the actual operation time when the DET puts down the import container in the buffer bracket in front of the import container area, and Equation (40) defines the actual operation time when the DET picks up the export container in the buffer bracket in front of the export container area.

Equations (41) and (42) define the specific waiting time of the DET if it arrives quayside too early when picking up the import container or delivering the export container. Similarly, when the DET is transporting the import container or picking up the export container, the buffer bracket is full, it also needs to wait, and the specific waiting time is shown in Equations (43) and (44), respectively.

Equations (45)–(48) define the continuous operation time constraint of DET corresponding to four different continuous container tasks, respectively: (i) continuously transport two import containers; (ii) continuously transport two export containers; (iii) first transport the import containers, then transport export containers; (iv) first transport export containers, then transport import containers.

Equations (49)–(51) define the variable type and value range, indicating that all the decision variables are binaries.

## 4. Example Application and Analysis

### 4.1. Task Description

There are 10 bays on the container ship, as shown in Figure 2. The number of containers that can be placed in each bay is limited by the number of rows and layers. The capacity is calculated according to the number of 8 layers of 18 rows with full-loads. The stowage diagram of each bay is randomly generated by Excel, as shown in Figure A1 in Appendix A.

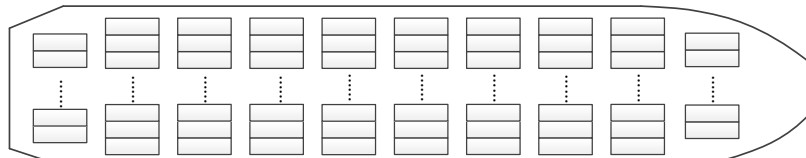

**Figure 2.** Schematic diagram of berthing container ship.

The handling process of "DTQC + DET + ARMG" is adopted in the terminal handling operation. The transport speed, operation energy consumption and other relevant parameters of quay crane and DET are shown in Table A4 in Appendix A. There is an import container area and an export container area in the yard, respectively, for stacking the containers unloaded/loaded by this ship.

### 4.2. Solution Strategy

As discussed in Section 2, the proposed 3 models are integrated by order. MODEL 1 optimize the loading and unloading sequences of the containers in the bay, achieving the minimum total quay crane operation time. Then, these optimized sequences will be set as the input information for MODEL 2, which optimizes the quay crane configuration and scheduling, with the objective of minimum total energy consumption for the quay crane operation. Next, based on the sequences of loading and unloading, as well as the quay crane configuration and scheduling, MODEL 3 is able to provide the optimum DET scheduling, in order to achieve minimum total energy consumption for the DET dispatching process.

MODEL 1 intent to optimize the loading and unloading sequence, which turns out to be a series of integer positives. Genetic Algorithm (GA) is selected as the methodology to work out the optimization. The detailed process is shown below.

(1) Coding method and initial population generation method

Each chromosome represents the loading and unloading sequence of the row of one bay. The negative number represents the unloading operation of each row in the bay, and the positive number represents the loading operation of each row in the bay.

The coding method is as follows: the unloading sequence of each row in the bay is generated randomly, and each digit of the chromosome represents the serial number of the row in the bay. Figure 3 shows the unloading sequence of 10 containers in this bay.

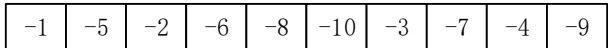

**Figure 3.** Coding graph of GA.

Then, loading sequence is added. After each unloading operation row number, traverse all the loading operation row numbers that can be inserted, select the row with the smallest difference from the number of containers to be unloaded in the target row, and insert the row number after the unloading operation row number, as shown in Figure 4.

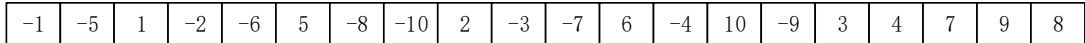

**Figure 4.** Example of container synchronous loading and unloading genetic algorithm coding.

In this example, six pairs of synchronous loading and unloading pairs were generated in this site, which were $(-5,1)$, $(-6,5)$, $(-10,2)$, $(-7,6)$, $(-4,10)$, $(-9,3)$.

(2) Fitness function

By optimizing the sequence of container loading and unloading in each bay, the number of synchronous loading and unloading of quay crane should be as many as possible. Therefore, the fitness function is set to the negative value of the objective function, so that the lower value of objective function leads to higher fitness.

(3) Chromosome selection

Chromosome selection was performed as follows:

① $N$ individuals ($N < M$, $N$ is the number of chromosomes in the current population) were randomly selected from the population, and the individuals with the largest fitness values were inherited to the next generation.

② The above process was repeated $Q$ times, one chromosome was selected each time, and a total of $Q$ chromosomes in the next generation population were obtained. The $Q$ chromosome is directly passed on to the next generation.

③ For the remaining $M$-$Q$ individuals, the roulette model is used to select.

(4) Genetic manipulation

The chromosomes in the population were crossed as follows. Select any two chromosomes, remove the loading column from the chromosome, select any same n row (n is selected according to the number of row s to be unloaded) from the two chromosomes, cross copy the n row in the two chromosomes, and obtain two new chromosomes without loading row. Then, according to the chromosome coding method, the loading sequence is added to the newly generated chromosome. The crossover method is shown in Figure 5: first, select $-1$, $-2$, $-3$ of the two chromosomes for crossover, and then add the loading row after the row of the newly generated offspring.

The chromosome was mutated as follows. Choose any two unloading positions of chromosome to exchange their positions. If each container is unloaded first and then loaded, the chromosome is feasible. If a container is loaded first and then unloaded, the chromosome of the unloading position in the mutated chromosome is adjusted to ensure that each container is unloaded first and then loaded. The crossover method is shown in Figure 6.

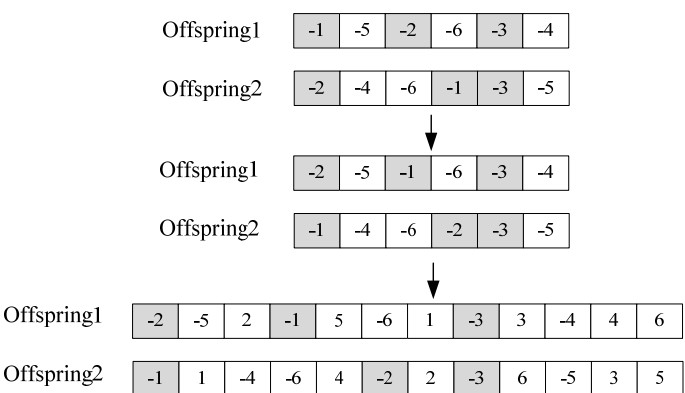

**Figure 5.** Crossover operation.

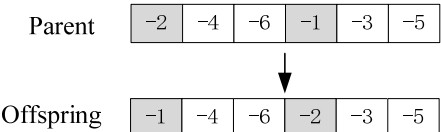

**Figure 6.** Mutation operation.

For Model 2, the scheduling scheme of quay crane is related to the position and moving direction of each quay crane within each time period, which is countable for a container ship to be loaded and unloaded, so that Enumeration Algorithm (EA) is utilized for solving exact solutions, is comprehensive.

As for the MODEL 3, finally, it is a mixed integer nonlinear programming problem, which results in difficulties in finding the accurate optimal solution. As a result, similar to the MODEL 1, GA serves as a proper alternative; meanwhile, the Simulated Annealing (SA) algorithm is introduced to obtain the initial solution for GA, in order to prevent the solutions falling into the local optimal. The detailed process of GA is similar with MODEL 1.

### 4.3. Optimization Results

#### 4.3.1. Loading and Unloading Sequence in Bay

According to the stowage diagram of different bays in Figure A1, MODEL 1 was solved by using MATLAB. The sequence of loading and unloading rows of 10 bays and the total operating time of each bay are shown in Table 1.

Taking bay 1 as an example, the loading and unloading sequence Gantt chart is shown in Figure 7. Within 0–10 min, the quay crane performs the unloading operation of the first row in a single cycle, and after the container in the first row is unloaded, it would carry out simultaneous loading and unloading operations. After 248 min, the 11th row will be loaded in a single cycle, and it will take 262 min to complete the container loading and unloading of all rows in bay 1.

**Table 1.** Sequence of loading and unloading of containers in different bay positions.

| Bays | Loading and Unloading Sequence | Total Operate Time/min |
|---|---|---|
| B1 | U1→U2-L1→U17-L2→U18-L17→U4-L18→U14-L4→U15-L14→U6-L15→U3-L6→U16-L3→U5-L16→U7-L5→U9-L7→U8-L9→U12-L8→U10-L12→U13-L10→U11-L13→L11 | 262 |
| B2 | U1→U2-L1→U9-L2→U4-L9→U5-L4→U7-L5→U8-L7→U11-L8→U12-L11→U15-L12→U3-L5→U10-L3→U16-L10→U17-L16→U18-L17→U6-L18→U14-L6→U13-L14→L13 | 276 |

**Table 1.** *Cont.*

| Bays | Loading and Unloading Sequence | Total Operate Time/min |
|---|---|---|
| B3 | U1→U2-L1→U9-L2→U7-L9→U3-L7→U4-L3→U16-L4→U10-L16→U11-L10→U5-L11→U8-L5→U12-L8→U14-L12→U6-L14→U13-L6→U15-L13→U17-L15→U18-L17→L18 | 270 |
| B4 | U1→U2-L1→U9-L2→U3-L9→U5-L3→U6-L5→U4-L6→U8-L4→U11-L8→U12-L11→U14-L12→U7-L14→U16-L7→U10-L16→U17-L10→U18-L17→U13-L18→U15-L13→L15 | 270 |
| B5 | U1→U2-L1→U9-L2→U4-L9→U5-L4→U15-L5→U6-L15→U7-L6→U8-L7→U11-L8→U12-L11→U16-L12→U10-L16→U17-L10→U18-L17→U14-L18→U3-L14→U13-L3→L13 | 274 |
| B6 | U1→U2-L1→U9-L2→U3-L9→U10-L3→U17-L10→U18-L17→U4-L18→U5-L4→U6-L5→U7-L6→U11-L7→U12-L11→U16-L12→U8-L16→U13-L8→U14-L13→U15-L14→L15 | 286 |
| B7 | U1→U2-L1→U8-L2→U3-L8→U9-L3→U10-L9→U17-L10→U18-L17→U4-L18→U7-L4→U11-L7→U12-L11→U6-L12→U13-L6→U14-L13→U16-L14→U15-L16→U5-L15→L5 | 278 |
| B8 | U1→U2-L1→U9-L2→U10-L9→U11-L10→U4-L11→U7-L4→U5-L7→U6-L5→U12-L6→U8-L12→U13-L8→U14-L13→U15-L14→U3-L15→U16-L3→U17-L16→U18-L17→L18 | 274 |
| B9 | U1→U2-L1→U9-L2→U4-L9→U6-L4→U7-L6→U3-L7→U8-L3→U12-L8→U5-L12→U14-L5→U15-L14→U13-L15→U16-L13→U10-L16→U11-L10→U17-L11→U18-L17→L18 | 276 |
| B10 | U1→U2-L1→U17-L2→U18-L17→U5-L18→U6-L5→U3-L6→U9-L3→U7-L9→U10-L7→U8-L10→U4-L8→U11-L4→U13-L11→U16-L13→U15-L16→U14-L15→U12-L14→L12 | 252 |

Note: U1 indicates that only the first row is unloaded, U2-L1 indicates that the second row is unloaded and the first row is loaded simultaneously, and L11 indicates that only the 11th row is loaded.

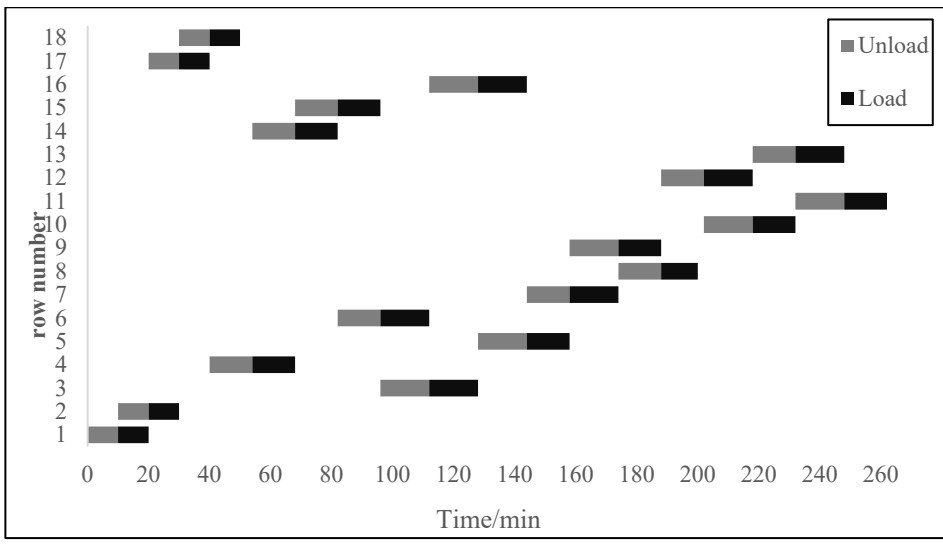

**Figure 7.** The loading and unloading sequence Gantt chart in bay 1.

### 4.3.2. Quay Crane Configuration and Scheduling

The total loading and unloading time window of a ship call in the port is 20 h, within which the terminal needs to dispatch the quay cranes to complete all loading and unloading operations. With the utilization of MATLAB again, MODEL 2 offers an optimized solution of the quay crane scheduling which is shown in Table 2.

**Table 2.** Computational result of quay crane dispatching schemes.

| Number of Quay Cranes | Total Operate Time/min | Energy Consumption/kWh | Sequence of Operation Bays | | Start-Finish Time |
|---|---|---|---|---|---|
| 1 | 2827 | 2,577,456 | Q1 | B1-B2-B3-B4-B5-B6-B7-B8-B9-B10 | 0–2827 min |
| 2 | 1416 | 257,676 | Q1 | B1-B2-B3-B4-B5 | 0–1410 min |
| | | | Q2 | B6-B7-B8-B9-B10 | 0–1416 min |
| 3 | 1125 | 257,606 | Q1 | B1-B2-B3 | 0–846 min |
| | | | Q2 | B4-B5-B6 | 0–854 min |
| | | | Q3 | B7-B8-B9-B10 | 0–1125 min |
| 4 | 836 | 258,081 | Q1 | B1-B2-B3 | 0–846 min |
| | | | Q2 | B4-B5 | 0–574 min |
| | | | Q3 | B6-B7 | 0–579 min |
| | | | Q4 | B8-B9-B-10 | 0–836 min |

### 4.3.3. DET Scheduling

According to the stowage diagram of container ships in Figure A1, it can be seen that a total of 2545 containers need to be moved by DETs. Based on the results of quay crane scheduling optimization, MATLAB programming is utilized to work out the different DET configuration schemes. In the scheduling scheme, the population size is set to 120, the maximum number of iterations is 1000, the initial temperature of the simulated annealing algorithm part is 1000, and the temperature drop rate r = 0.98. As a result, the reasonable DET resource to complete the designed job fits in the interval between 15 and 21, while their corresponding time and energy consumption are shown in Table 3 correspondingly.

**Table 3.** Results of different configuration schemes of DETs.

| Number of DETs | Total Operation Time/h | Energy Consumption/kWh |
|---|---|---|
| 15 | 25.81 | 11,614.52 |
| 16 | 23.71 | 11,380.83 |
| 17 | 21.93 | 11,184.31 |
| 18 | 20.92 | 11,296.82 |
| 19 | 19.98 | 11,388.64 |
| 20 | 19.33 | 11,598.01 |
| 21 | 18.94 | 11,932.23 |

### *4.4. Result Discussions*

#### 4.4.1. Analysis of Optimization Results

As mentioned in Section 4.3, through the optimization model, the integrated scheduling plan for the "DTQC + DET + ARMG" system could be obtained, indicating the feasibility of the proposed model.

On the basis of the optimized loading and offloading sequence in the bay (as shown in Section 4.3.1), the quay crane configuration and scheduling are achieved. As can be seen in Table 2, when the number of involved quay cranes less than three, the minimum time

consumption is 1416 min, which may exceed the maximum time window allowance of 20 h. The energy consumption of three quay cranes is similar to that of two but fulfill the requested time window. However, when the number of involved quay cranes increase to four, there will not be any challenge to meet the operational time requirement but the total energy consumption will be much higher than before. Therefore, the optimal solution of quay cranes employment comes to three with a time requirement of 1125 mins at the energy consumption of 257,606 kWh. The optimal scheduling under this configuration is: quay crane 1 operates B1-B3 in sequence, quay crane 2 operates B4-B6 in sequence, and quay crane 3 operates B7-B10 in sequence.

Using the optimized quay crane configuration and scheduling results, the optimal DET scheduling is obtained, as shown in Table 3. DET scheduling. Results shows that the allocated job cannot be performed in 20 h with the inputs of 18 DETs or less. Along with the continuous increase in DET involvement, the overall time required shows a clear decreased tendency and all meet the time window requirement, even though the energy consumption becomes higher and higher. As a result, the most appropriate solution comes from the 19 DETs involvement in the 19.98 hours operation at an energy consumption of 11,388.64 kWh. In addition, the proposed model is flexible to different scenarios which adopt different types of trucks. Replacing the DET in the original model by manual-driving diesel trucks or AGVs, the corresponding integrated scheduling plan could also be optimized. According to these optimization results, the comparison of different horizontal transportation equipment is then conducted from the perspective of operation efficiency, economy and sociality, respectively.

### 4.4.2. Comparison with Traditional Horizontal Transportation Vehicles

(1) Operation efficiency analysis

The allocation of those three types of horizontal transportation vehicles ranges from 15 to 25, the total corresponding operation time and operation efficiency are shown in Table 4 and Figure 8.

**Table 4.** Comparison of operation system efficiency.

| Quantity | Manual Driving Diesel Truck | | AGV | | DET | |
|---|---|---|---|---|---|---|
| | Operation Time (h) | Operation Efficiency (TEU·h) | Operation Time (h) | Operation Efficiency (TEU·h) | Operation Time (h) | Operation Efficiency (TEU·h) |
| 15 | 28.31 | 89.90 | 31.71 | 80.26 | 25.81 | 98.61 |
| 16 | 26.31 | 96.73 | 29.41 | 86.54 | 23.71 | 107.34 |
| 17 | 24.51 | 103.84 | 27.41 | 92.85 | 21.93 | 116.05 |
| 18 | 23.01 | 110.60 | 25.71 | 98.99 | 20.92 | 121.65 |
| 19 | 21.81 | 116.69 | 24.21 | 105.12 | 19.98 | 127.38 |
| 20 | 21.01 | 121.13 | 22.91 | 111.09 | 19.33 | 131.66 |
| 21 | 20.51 | 124.09 | 21.96 | 115.89 | 18.94 | 134.37 |
| 22 | 20.21 | 125.93 | 21.07 | 120.79 | 18.66 | 136.39 |
| 23 | 19.88 | 128.02 | 20.4 | 124.75 | 18.45 | 137.94 |
| 24 | 19.68 | 129.32 | 20.09 | 126.68 | 18.27 | 139.30 |
| 25 | 19.54 | 130.25 | 19.87 | 128.08 | 18.13 | 140.38 |

Within the given interval of horizontal transportation vehicles allocation between 15 and 25, the overall time consumptions of all those three types of vehicles shows a clear decreased tendency, while the corresponding operational efficiencies are consequently improved. Meanwhile, it can also be observed that the improvement of operational efficiency, or the saving of overall operational time consumption, becomes less significant once the allocated number of vehicles is greater than the threshold value of 19. It indicates a clear marginal utility within the process of vehicle allocation.

In regard to the individual vehicles, DET and AGV hold the most and least significant efficiency advantage, respectively, while the manual driving diesel truck is set in the middle. As the maximum travel speed of DET can reach about 35 km/h and it will only take about

5 min to replace the battery for a non-stopping operation. The manual driving diesel truck may run at the same travel speed, but the work shift cannot be as seamless as the DET. AGV's designed travel speed in the port area is only about 20 km/h despite its great advantage in safety and labor saving. For example, given the task of loading and discharging containers of 2545 within the time window of 20 h, the request vehicles are 19 of DET, 25 of AGV and 23 of manual driving diesel trucks, respectively.

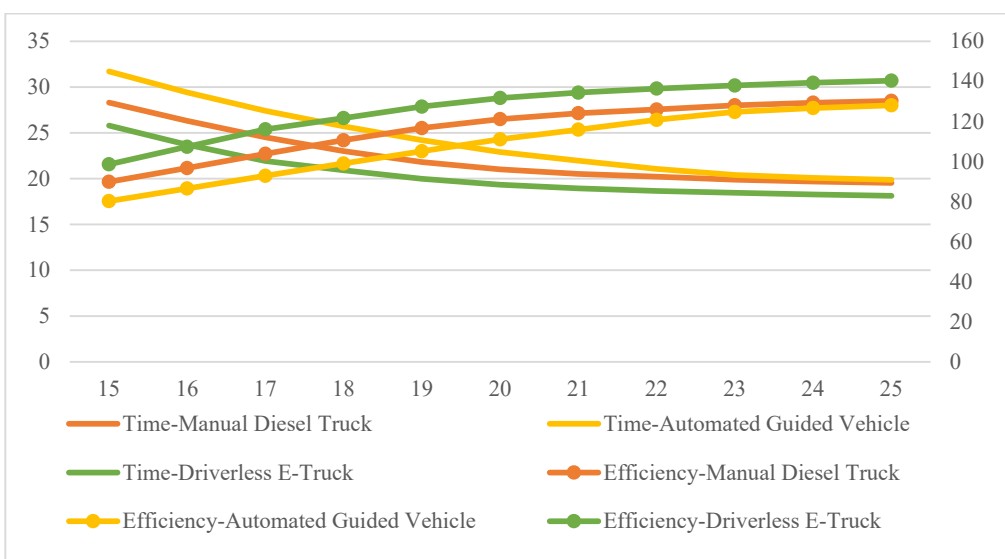

**Figure 8.** Comparison of horizontal transportation vehicles.

(2) Economic analysis

Taking the same task of 2545 container movement within the time window of 20 h, the overall transport costs of those three corresponding horizontal transport vehicles are computed by taking into account of the components of vehicle purchasing, tax, insurance, labor, fuel, etc. Please see Figure 9 below for the results while Table A3 in Appendix A shows specific cost components and corresponding calculation methods.

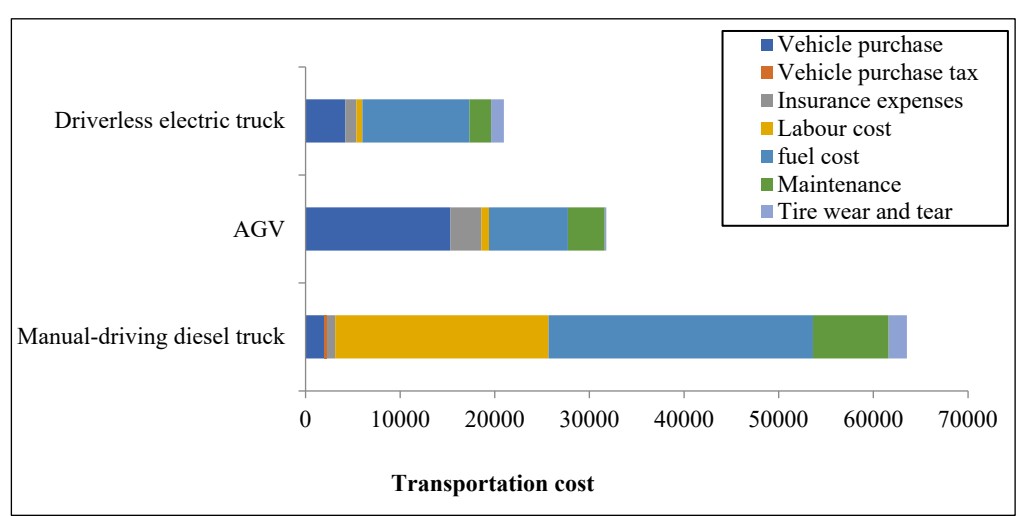

**Figure 9.** Comparison of transportation costs of three horizontal transportation vehicles.

Results indicate that the traditional manual driving diesel truck has been the most expensive option, costing about 64,000 RMB to move 2545 containers. Without any doubts, the major costs come from the fuel and labor section, about 43.98% and 35.48% of overall costs, respectively. The AGV spent only about half of the cost of the manual driving diesel

vehicle to complete the task. However, it is worth noting that vehicle cost has been one of the main disadvantages of AGV's application, in which 48.17% of its overall cost comes from the vehicle purchasing. The DET is the most economical choice as it only cost about 22,000 RMB for the given task, which is about two thirds of the AGV's and one third of the traditional manual driving diesel truck. This is mainly due to the significant saving in vehicle purchasing and labor costing. It is particularly important nowadays as the shortage of truck driver becomes more and more serious.

In addition, in the case studies, the input parameters, for instance, the energy consumptions of DET and AGV in different operation processes, etc., are selected based on current typical situations, in order to provide a basic data set for presenting the application of the proposed models. Considering the development trend of DET and AGV will be toward lower energy consumption and lower cost. As a result, a detailed sensitively analysis for the results of the proposed models makes sense in relation to the future development of DET and AGV, which will be one aspects of our future work.

## 5. Conclusions

In this paper, the integrated scheduling optimization of the container handling system "dual-trolley quay crane + driverless electric truck (DET) + (Automatic rail mounted gantry) ARMG" was proposed. In such system, the driverless electric container truck, a new type of horizontal transportation equipment, is introduced. To conduct the integrated optimization, three models were developed, including the optimization of the loading and unloading sequence in bay (MODEL 1), optimization of quay crane configuration and scheduling (MODEL 2) and optimization of DET scheduling (MODEL 3). Genetic Algorithm was applied to solve MODEL 1 and MODEL 3, while the Enumerated Algorithm was used for MODEL 2. In addition, simulated annealing was combined with GA in the solution of MODEL 3 in order to prevent the results from being trapped into local optimal.

Through the optimization, the container loading and unloading plan, the corresponding configuration and scheduling of quay crane, and the scheduling of DET for the containers' horizontal transportation could be figured out, which minimize the container loading and unloading operation time, and the total energy consumption for quay cranes and DETs. The case studies indicate the feasibility of the proposed models and algorithms, at the same time, through case studies, the performance of the newly introduced DET, was compared with the common manual driving diesel truck and AGV. DET turns out to be a better alternative due to lower cost, lower energy consumption as well as the reduction in labor costs.

Furthermore, there are still some issues that can be further discussed in the future.

(1) The container sizes are assumed to be the same. So, for the future work, the scheduling optimization problem of loading and unloading equipment for the mixed loading of 20 ft and 40 ft containers, refrigerated containers and dangerous goods containers will be with more practical value.

(2) In this paper, only the static scheduling process was considered, leaving out the fault of the quay crane and DET, path conflict and path congestion in real-time road conditions. Therefore, the real-time dynamic scheduling problem can be further studied in the future.

**Author Contributions:** Conceptualization, Y.W.; methodology, Y.W. and Y.G.; validation and formal analysis, Y.W., Y.G., C.H. and T.L.; writing—original draft preparation, Y.G. and Y.W.; writing—review and editing, C.H., Y.W. and T.L.; funding acquisition, Y.W. and C.H. All authors have read and agreed to the published version of the manuscript.

**Funding:** This research was funded by Natural Science Foundation of Ningbo, grant number 2021J111, Natural Science Foundation of Zhejiang Province, grant number LQ23E080011, Natural Science Foundation of Ningbo, grant number 2022J088, and Natural Science Foundation of Zhejiang Province, grant number LQ21G010001.

**Institutional Review Board Statement:** Not applicable.

**Informed Consent Statement:** Not applicable.

**Data Availability Statement:** Not applicable.

**Conflicts of Interest:** The authors declare no conflict of interest.

## Appendix A

**Table A1.** Sets, parameters, and variables for MODEL 1.

| **Sets** | |
| --- | --- |
| $Q$ | Set of quay cranes, $Q = \{1, 2, \cdots, q\}$ |
| $B$ | Set of bays, $B = \{1, 2, \cdots, b\}$ |
| $R$ | Set of rows, $R = \{1, 2, \cdots, r\}$ |
| **Parameters** | |
| $N_{br}^u$, $N_{br}^l$ | Number of containers to be unloaded and loaded in row $r$ of bay $b$, respectively |
| $s_{br}^u$, $e_{br}^u$ | The start time and end time of unloading the container in row $r$ of bay $b$ |
| $s_{br}^l$, $e_{br}^l$ | The start time and end time of loading the container in row $r$ of bay $b$ |
| $T_1$ | Time for the main trolley of quayside bridge to complete one operation of taking/releasing containers |
| $T_b$ | Total operating time of bay $b$ |
| $T_{bq}$ | The completion time of quay crane $q$ loading and unloading all containers in bay $b$ |
| **Decision variables** | |
| $x_{rr'}$ | Binary variables, if row $r$ is unloaded before row $r'$, the value is 1, otherwise, 0 |
| $y_{rr'}$ | Binary variables, if row $r$ is loaded before row $r'$, the value is 1, otherwise, 0 |

**Table A2.** Sets, parameters, and variables for MODEL 2.

| **Sets** | |
| --- | --- |
| $\psi$ | Tasks that cannot be operated at the same time, $\psi = \{(b, b') \mid b, b' \in B, \lvert l_{b'} - l_b \rvert \le l_s\}$ |
| **Parameters** | |
| $l_b$ | Location of bay $b$ |
| $l_{sq}$ | The bay position of quay crane $q$ when it started operation |
| $l_{eq}$ | The bay position of quay crane $q$ when it ended operation |
| $l_s$ | Safety distance between bays |
| $T_2$ | The time for the quay crane to move one bay along the ship |
| $W_1$, $W_2$, $W_3$ | Energy consumption per unit time of each quay crane during Operation, Moving and Waiting, respectively |
| $T_F$ | Total handling time of container ships in port |
| $T_{bb'}$ | The waiting time for quay crane to finish container operation in Bay $b$ and then go to bay $b'$ |
| **Decision variables** | |
| $x_{bq}$ | Binary variables, if quay crane $q$ is loading and unloading at bay $b$, the value is 1, otherwise, 0 |
| $z_{bb'q}$ | Binary variables, if the operation in bay $b$ is in front of bay $b'$, the value is 1, otherwise, 0 |

**Table A3.** Sets, parameters, and variables for MODEL 3.

| Sets | |
|---|---|
| $A$ | Set of DETs, $A = \{1, 2, \cdots, a\}$ |
| $U, L$ | Set of import and export containers |
| $N$ | Set of all containers, $N = \{1, 2, \cdots, n\}$ |
| $C$ | Set of all ARMGs, $C = \{1, 2, \cdots, c\}$ |

| Parameters | |
|---|---|
| $T_{nq1}^{p}$ | Planned operation time of the main trolley of quay crane $q$ for the container $n$ |
| $T_{nq1}^{r}$ | Actual operation time of the main trolley of quay crane $q$ for the container $n$ |
| $T_{q1}^{p\max}$ | The time when the main trolley of quay crane $q$ plans to operate the last container |
| $T_{nq2}^{p}$ | Planned operation time of the gantry trolley of quay crane $q$ for the container $n$ |
| $T_{nq2}^{r}$ | Actual operation time of the gantry trolley of quay crane $q$ for the container $n$ |
| $T_{nq2}^{E}$ | The earliest time when the gantry trolley of the quay crane $q$ can pick up/put down the container $n$ on the transfer platform |
| $T_{nq2}^{L}$ | The latest time when the gantry trolley of the quay crane $q$ can pick up/put down the container $n$ on the transfer platform |
| $T_{nc}^{U}$ | The moment when the ARMG on the seaside of the import container area pick up the import container $n$ from the buffer bracket |
| $T_{nc}^{L}$ | The moment when the ARMG on the seaside of the export container area put down the export container $n$ from the buffer bracket |
| $T_{na}^{q}$ | The moment when the DET $a$ starts to operate container $n$ under the gantry trolley of quay crane $q$ |
| $T_{na}^{L}$ | The moment when the driverless electric container truck $a$ starts to operate container $n$ in the buffer bracket in front of the export container area |
| $T_{na}^{EU}$ | The earliest time when the driverless electric container truck $a$ can put down the container $n$ on the buffer bracket in front of the import container area |
| $T_{na}^{LU}$ | The latest time when the driverless electric container truck $a$ can put down the container $n$ on the buffer bracket in front of the import container area |
| $T_{na}^{EL}$ | The earliest time when the driverless electric container truck $a$ can pick up the container $n$ on the buffer bracket in front of the export container area |
| $T_{na}^{LL}$ | The latest time when the driverless electric container truck $a$ can pick up the container $n$ on the buffer bracket in front of the export container area; |
| $w_{naq}$ | Waiting time of the driverless electric container truck $a$ carrying container $n$ under quay crane $q$ |
| $w_{nac}$ | Waiting time of the driverless electric container truck $a$ carrying container $n$ in the yard buffer area |
| $T_3$ | The average time for the gantry trolley to complete one operation of taking/releasing containers |
| $T_4$ | The average time for ARMG to complete one operation of taking/releasing containers |
| $T_5$ | Battery swap time of each DET |
| $T_{na}^{qU}$ | The time taken for the DET $a$ to travel from the gantry trolley of quay crane $q$ to the area in front of the import container area |
| $T_{na}^{qL}$ | The time taken for the DET $a$ to travel from the gantry trolley of quay crane $q$ to the area in front of the export container area |
| $T_{na}^{Uq'}$ | The time taken for the DET $a$ to travel from the area in front of the import container area to the gantry trolley of quay crane $q'$ |
| $T_{na}^{Lq'}$ | The time taken for the DET $a$ to travel from the area in front of the export container area to the gantry trolley of quay crane $q'$ |
| $T_{a}^{UL}$ | The time taken for the DET $a$ to travel from the area in front of the import container area to the area in front of the export container area |
| $T_{a}^{qq'}$ | The time taken for the DET $a$ to travel from the gantry trolley of quay crane $q$ to the gantry trolley of quay crane $q'$ |
| $T_{nn'a}$ | The empty driving time of the next container after the container $n$ is finished by the DET $a$ |
| $T_{a}^{E}$ | Time from DET $a$ to charging station |
| $W_4$ | Waiting energy consumption per unit time of gantry trolley waiting for DET |
| $W_5$ | Energy consumption per unit time of full-load movement of each DET |
| $W_6$ | Energy consumption per unit time of no-load movement of each DET |
| $W_7$ | Waiting energy consumption per unit time of each DET |
| $p_1$ | Capacity of transfer platform of dual-trolley quay crane |
| $p_2$ | Capacity of the buffer bracket in the yard |
| $v_1$ | Driving speed of DET with full-load |
| $v_2$ | Driving speed of DET with no-load |
| $L_{\max}$ | Maximum range of each DET |

| Decision variables | |
|---|---|
| $y_{na}$ | Binary variables, if container $n$ is transported by driverless electric truck $a$, the value is 1, otherwise, 0 |
| $y_{nc}$ | Binary variables, if container $n$ is operated by ARMG $c$, the value is 1, otherwise, 0 |
| $z_{nn'a}$ | Binary variables, if the driverless electric truck $a$ transports container $n$ first and then transports container $n'$, the value is 1, otherwise, 0. |
| $u_a$ | Binary variables, if the remaining power of driverless electric truck $a$ is less than the safe power, the value is 1, otherwise, 0 |

**Table A4.** Driving parameters of DET.

| Parameters | Meaning | Value | Unit |
| --- | --- | --- | --- |
| $m_1$ | Self-weight of DET (tractor + Trailer) | 17 | t |
| $m_2$ | Standard container weight | 3.8 | t |
| $m_3$ | Standard container rated load | 20 | t |
| $\beta$ | Container loading factor | 0.7 | - |
| $g$ | Acceleration of gravity | 9.81 | m/s$^2$ |
| $c_{roll}$ | Rolling resistance coefficient | 0.01 | - |
| $c_{air}$ | Coefficient of air resistance | 0.695 | - |
| $A$ | Windward area of DETs | 9.6 | m$^2$ |
| $\rho_{air}$ | Air density | 1.2 | kg/m$^3$ |
| $\bar{i}$ | Average gradient during transportation | 0.062 | % |
| $n_{acc}$ | Average number of accelerations per kilometer | 0.1 | times/km |
| $\bar{v}_1$ | Average driving speed of DET with full-load | 30 | km/h |
| $\bar{v}_2$ | Average driving speed of DET with no-load | 35 | km/h |

**Table A5.** Equipment parameters of the terminal.

| Parameter | Value | Unit | Parameter | Value | Unit |
| --- | --- | --- | --- | --- | --- |
| $T_1$ | 2 | min | $l_s$ | 1 | bay |
| $T_2$ | 1 | min | $W_1$ | 91.24 | kWh/(h·veh) |
| $T_3$ | 1 | min | $W_2$ | 70.18 | kWh/(h·veh) |
| $T_4$ | 3 | min | $W_3$ | 49.6 | kWh/(h·veh) |
| $p_1$ | 2 | - | $W_4$ | 49.6 | kWh/(h·veh) |
| $p_2$ | 4 | - | $W_5$ | 34.05 | kWh/(h·veh) |
| $v_1$ | 30 | km/h | $W_6$ | 26.84 | kWh/(h·veh) |
| $v_2$ | 35 | km/h | $W_7$ | 13.62 | kWh/(h·veh) |

**Table A6.** Transportation cost composition.

| Transport Cost Component | | Detail | Calculation Formula |
| --- | --- | --- | --- |
| Fixed cost | Vehicle purchase | In the form of vehicle depreciation, it is apportioned within a certain period of time. | $C_{ci} = \frac{c_{bi} \times (1-r_i) \times T_i}{8760 n_i}$ |
| | Vehicle purchase tax | A vehicle purchase tax shall be levied on newly purchased vehicles, which shall be collected at a certain tax rate on the basis of the taxable price of the new vehicle. | $C_{ti} = \frac{c_{bi} \times 10\% \times T_i}{1.09 \times 8760 n_i}$ |
| | Insurance expenses | The types of motor vehicle insurance in my country are mainly divided into two categories: compulsory motor vehicle traffic accident liability insurance and commercial insurance. In this article, commercial insurance mainly considers vehicle loss insurance. | $C_{ii} = c_{ii}^1 + c_{ii}^2$ |
| | Labor cost | The traditional manual driving truck has the driver to drive the vehicle, and the unmanned vehicle has a dedicated operator to control the vehicle in the monitoring room. | $C_{si} = \frac{\bar{c}_{si} \times T_i}{8760}$ |
| Operating costs | Fuel cost | The fuel used in traditional manual driving trucks is diesel, while AGV and driverless trucks use pure electric form. The specific fuel consumption is related to the energy consumption of the vehicle. | $C_{fi} = f_i \times \bar{c}_{fi}$ |
| | Maintenance | When the vehicle transportation mileage reaches a certain distance, the corresponding maintenance work will be carried out, which is usually calculated according to yuan/(100 km·vehicle). | $C_{wi} = \frac{L_{Ri}}{L_{wi}} \times \bar{c}_{wi}$ |
| | Tire wear and tear | The tire has a certain life span. After reaching a certain transportation distance, the tire must be replaced, and the loss is calculated according to the mileage. | $C_{li} = \frac{n_{li} \times \bar{c}_{li} \times L_{Ri}}{L_{li}}$ |

Where $C_{ci}$ is the purchase cost of vehicles with different kinds of transportation equipment, yuan, when $i = 1$ is manually driven diesel truck, $i = 2$ is AGV, $i = 3$ is DET; $C_{bi}$ is the cost of purchasing different kinds of vehicle, yuan; $r_i$ is the residual value rate of

the vehicle after the end of the depreciation period, %; $n_i$ is the depreciation period of the vehicle, year; $T_i$ is the transport time, h; $C_{ti}$ is the vehicle purchase tax for different kinds of vehicles, yuan; $C_{ii}$ is the vehicle insurance premium for different kinds of vehicles, yuan; $c_{ii}^1$ is compulsory insurance for motor vehicle accident liability, yuan; $c_{ii}^2$ is the vehicle loss insurance, yuan; $C_{si}$ is the labor cost when using different kinds of vehicles, yuan; $\overline{c_{si}}$ is the average annual salary of the driver/operator, yuan; $C_{fi}$ is the cost of fuel consumed by different kinds of vehicles, yuan; $f_i$ is fuel consumption; $\overline{c_{fi}}$ is the average price per unit of fuel; $C_{wi}$ is the maintenance fee for different kinds of vehicles, yuan; $L_{Ri}$ is transport distance, km; $L_{wi}$ is the rated distance for corresponding maintenance work, km; $\overline{c_{wi}}$ is the average cost of each repair and maintenance, yuan; $C_{li}$ is the tire wear and tear costs of different levels of transportation equipment, yuan; $n_{li}$ is the number of tires of different kinds of vehicles; $\overline{c_{li}}$ is the average price of a tire; $L_{li}$ is the distance that the tire can travel during its life, km.

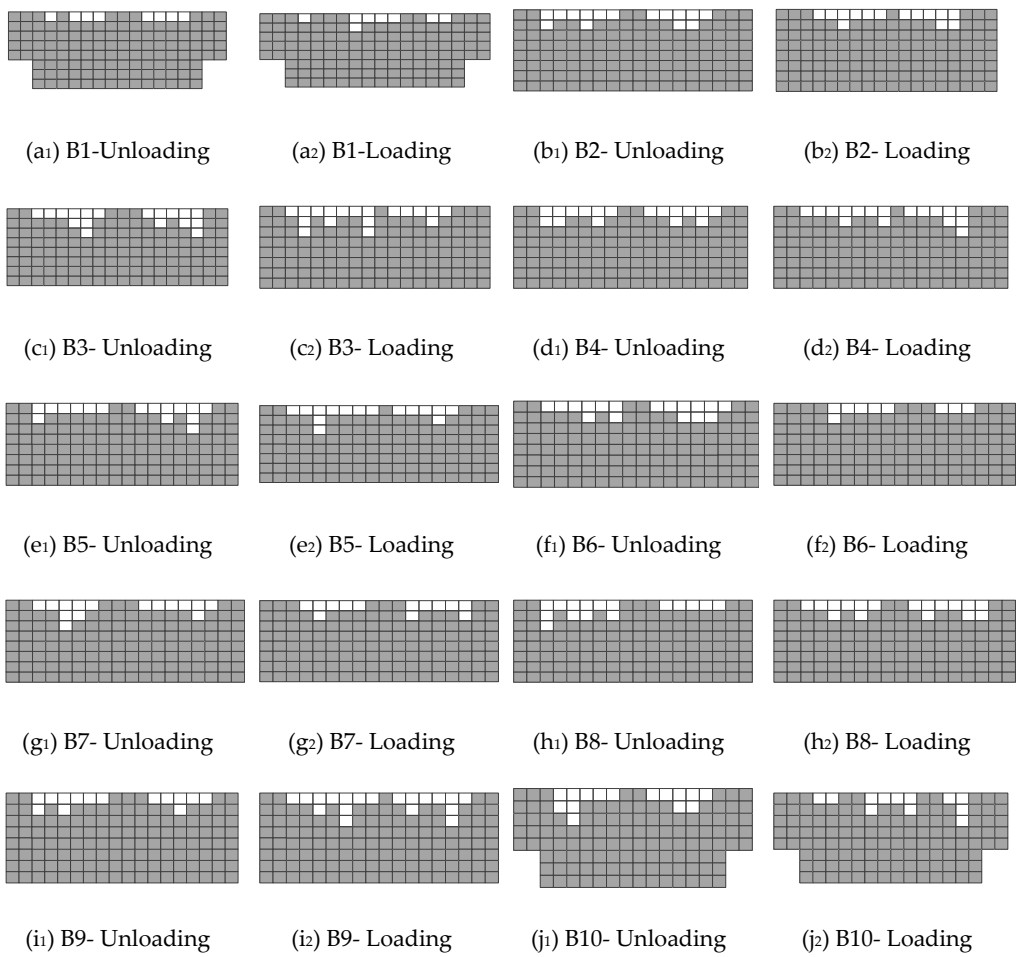

(a₁) B1-Unloading    (a₂) B1-Loading    (b₁) B2- Unloading    (b₂) B2- Loading

(c₁) B3- Unloading    (c₂) B3- Loading    (d₁) B4- Unloading    (d₂) B4- Loading

(e₁) B5- Unloading    (e₂) B5- Loading    (f₁) B6- Unloading    (f₂) B6- Loading

(g₁) B7- Unloading    (g₂) B7- Loading    (h₁) B8- Unloading    (h₂) B8- Loading

(i₁) B9- Unloading    (i₂) B9- Loading    (j₁) B10- Unloading    (j₂) B10- Loading

**Figure A1.** Stowage diagram for each bay of container ship.

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
