# Peer review of "The Integrated Scheduling Optimization for Container Handling by Using Driverless Electric Truck in Automated Container Terminal"

_sustainability, doi:10.3390/su15065536_

Round 1

Reviewer 1 Report

1. The cited reference should be improved add more references from the recent years

2. Please check the use of words/grammar in the paper by using an advanced English checker or application i.e., Grammarly.

Reviewer 2 Report

1. The background of the research is not comprehensively stated in the introduction.

2. The purpose and innovation of the research are not well defined.

3. The structure of this paper can be improved

4. In the section of Theoretical foundations, authors can draw a diagram to demonstrate the theoretical framework.

5.Does your research help increase global knowledge?

6. What problems does your research solve for the target community?

7. What is the difference between your research and other researches? Compare your research with others.

Reviewer 3 Report

This paper to explore the configuration and optimized scheduling model of the driverless electric container truck with an objective of minimizing overall energy consumption.This paper proposes the integrated scheduling optimization of the container handling system “dual-trolley quay crane + driverless electric truck (DET) + (Automatic rail mounted gantry) ARMG”.

This paper deals with an important subject, it is very interesting, well organized. However, some shortcomings should be overcome in the manuscript. 

Some comments are as follows:

(1) In the article, model 1 is resolved by genetic algorithm, but the article does not introduce the application of genetic algorithm in detail, nor does it give the advantages of genetic algorithm in model 1.

(2) This paper uses model 1,2 and 3 to realize the scheduling optimization problem. In the actual process, whether the three models will interfere and affect each other? What is the basis for determining the objective functions of the three models?It is not explained in the article.

(3) Some abbreviations do not have full names,for example: AGV and so on.

(4)The tables in the text should be in the format required by the journal (three-line table). Figure 1 is ambiguous, and it is recommended to replace it.

(5) There are a large number of variables in the article(Table1-3). It is recommended to put the variables in Appendix.

Reviewer 4 Report

The paper focuses on planning loading, unloading and equipment scheduling activities at an automated port. There are some contributions. However, I have several major concerns on this paper. If these major concerns are not addressed in a revision, I intend to ask rejection in the next round:

1- The contribution statement of this paper between line 72-83 is badly written and not clear. You should rewrite that paragraph.

Additionally, in line 72-83, authors claim that there is no study integrating loading, unloading and horizontal transport operations. This is a very bad claim. There are so many papers doing that. This means authors did not investigate the literature properly. You may be the first one focusing on driverless electric trucks. You can emphasise that.

2- To me, the difference between an AGV and DET is not clear at all. There might not be any difference between AGV and DET. AGVs are also driverless (central control). AGVs are also electrified. You need to clearly explain differences in the introduction.

3- The following relevant study optimises the loading planning and equipment scheduling in an integrated way. I encourage you to read and cite the work in your study:

2018. Flexible ship loading problem with transfer vehicle assignment and scheduling. Transportation Research Part B: Methodological, 111, pp.113-134.

4- I really have concerns on the correctness of the formulations. In the literature, there are several papers with published/working models. Authors could have solved an already available formulation with your new equipment. Adding new formulations to the literature is not best option.

5- In the set definitions, there are bays and rows. However, tiers are missing.

The starting and ending time of (un)loading is presented as a parameter. However, these are normally decision variables. How can you know loading start time if you have sequencing as a decision variable? This seems wrong modelling.

6- x_bq variable relates to quay crane (un)loading at bay b. However this bay is on the ship. I assume the actual bay definition is in table 1 is on the yard. These two bays are completely different. for x_bq, you have to define new bays for the vessel.

7- section 4.2. is very short. There are no details about the solution algorithm. This reduces rigor of this study.

8- Authors mention about energy consumption and equipment planning. Following studies focus on the topic. Authors can cite and discuss in the paper:

2019. A review of energy efficiency in ports: Operational strategies, technologies and energy management systems. Renewable and Sustainable Energy Reviews, 112, pp.170-182.

2021. Optimal energy management and operations planning in seaports with smart grid while harnessing renewable energy under uncertainty. Omega, 103, p.102445.

Reviewer 5 Report

Section 1: Literature review can be improved. There exist recent works on port seaside operations management regarding ship service. Please include the following related works to improve the comprehensiveness of the literature review:

1. Equitable vessel traffic scheduling in a seaport. Transportation Science, 56(1):162–181.

2. A simulation optimization method for deep-sea vessel berth planning and feeder arrival scheduling at a container port. Transportation Research Part B, 142:174–196.

Section 2: Research problem is properly presented. Please also add a discussion on the differences between the operations of DET and tradditional trucks. The advantage of DET should be highlighted.

Section 3: Models are adequately designed. However, the relationship between the three models and how to integrate the three models to generate an overall service plan need to be further clarified. 

Section 4: Computational results are adequately reported. 

Section 5: Concludion part is adequate.

Round 2

Reviewer 4 Report

I appreciate authors' effort to address my comments. Most of my comments are well addressed, but still I have major concerns on following bullets. The paper is progressing well, but the paper needs another major revision.

1- In the response file authors note that "Besides, in MODEL 1, there is no constraints indicating the relation between starting/ending time of loading/unloading operations and the operation sequences, this is due to the applied solution method, genetic algorithm. When applying genetic algorithm, the process of genenrating candidate solution integrates the squences of loading/unloading operations and the starting/ending times."

This statement shows that there is a problem with the model. Your model must reflect actual relationships in the operations. Therefore, there must be constraints setting starting and ending times in the model. Your solution method GA can fix some decisions but your model does not have to know about this. Model and algorithm are two seperate entities. your model should be improved.

2- I encourage authors to conduct further sensitivity analyses to show the power of the results. You should report results for changes in the input parameters.

3- Line 36 "against" should be "due to"

Line 95-96; following studies can be cited and added to the list: [1-2]

[1] 2019. Recoverable robustness in weekly berth and quay crane planning. Transportation Research Part B: Methodological, 122, pp.365-389.

[2] 2015. Integrated berth allocation and quay crane assignment problem: Set partitioning models and computational results. Transportation Research Part E: Logistics and Transportation Review, 81, pp.75-97.

Line 100; presents should be presented

Reviewer 5 Report

The authors have adequately addressed my comments. I would like to recommend acceptance of the paper.

Author Response

The authors would like to thank the reviewer for his time, comments and fruitful suggestions.

Round 3

Reviewer 4 Report

The model 1 has changed but computational remains the same. I hope authors have validated results.

Authors should update results in production stage if it is required.

Author Response

Thanks for the comment! The results have already been validated.

When applying GA to solve the model, the candidate solutions are generated according to Eq. (6) and Eq. (7). Actually, these two sets of constraints had already been considered in the original version, though without explicitly mentioned in the first submitted manuscript. Therefore, the results remain the same in the revised manuscript, and Eq. (6) and Eq. (7) are included to make the proposed model more complete.